# Using ChEMBL to Complement Schistosome Drug Discovery

**DOI:** 10.3390/pharmaceutics15051359

**Published:** 2023-04-28

**Authors:** Gilda Padalino, Avril Coghlan, Giampaolo Pagliuca, Josephine E. Forde-Thomas, Matthew Berriman, Karl F. Hoffmann

**Affiliations:** 1School of Pharmacy and Pharmaceutical Sciences, Cardiff University, Redwood Building, King Edward VII Avenue, Cardiff CF10 3NB, UK; 2Wellcome Sanger Institute, Wellcome Genome Campus, Cambridge CB10 1SA, UK; alc@sanger.ac.uk; 3Independent Researcher, Cardiff CF3 3LT, UK; giampaolo.pagliuca@outlook.com; 4The Department of Life Sciences (DLS), Aberystwyth University, Aberystwyth SY23 3DA, UK; jef19@aber.ac.uk; 5Wellcome Centre for Integrative Parasitology, School of Infection and Immunity, University of Glasgow, 120 University Place, Glasgow G12 8TA, UK; matt.berriman@glasgow.ac.uk

**Keywords:** ChEMBL, schistosomiasis, drug discovery pipeline, schistosomula, adult worm, bioinformatics, cytotoxicity

## Abstract

Schistosomiasis is one of the most important neglected tropical diseases. Until an effective vaccine is registered for use, the cornerstone of schistosomiasis control remains chemotherapy with praziquantel. The sustainability of this strategy is at substantial risk due to the possibility of praziquantel insensitive/resistant schistosomes developing. Considerable time and effort could be saved in the schistosome drug discovery pipeline if available functional genomics, bioinformatics, cheminformatics and phenotypic resources are systematically leveraged. Our approach, described here, outlines how schistosome-specific resources/methodologies, coupled to the open-access drug discovery database ChEMBL, can be cooperatively used to accelerate early-stage, schistosome drug discovery efforts. Our process identified seven compounds (fimepinostat, trichostatin A, NVP-BEP800, luminespib, epoxomicin, CGP60474 and staurosporine) with ex vivo anti-schistosomula potencies in the sub-micromolar range. Three of those compounds (epoxomicin, CGP60474 and staurosporine) also demonstrated potent and fast-acting ex vivo effects on adult schistosomes and completely inhibited egg production. ChEMBL toxicity data were also leveraged to provide further support for progressing CGP60474 (as well as luminespib and TAE684) as a novel anti-schistosomal compound. As very few compounds are currently at the advanced stages of the anti-schistosomal pipeline, our approaches highlight a strategy by which new chemical matter can be identified and quickly progressed through preclinical development.

## 1. Introduction

Schistosomiasis is a neglected tropical disease caused by infection with blood fluke schistosomes. It causes up to 13,000 human deaths per year [1], increases the risk of developing certain types of cancers, skews the immune system in infected individuals leading to reduced efficacy of vaccines, drives the development of genital lesions in both males and females and is responsible for millions of disability adjusted life years lost in affected communities [2]. For decades, schistosomiasis control in individuals, communities and country-wide mass drug administration programs has been predominantly driven by treatment with drugs such as praziquantel (current frontline drug globally used against all *Schistosoma* species [3]) and oxamniquine (historically used in Brazil against *Schistosoma mansoni* [4]). The combinatorial application of interdisciplinary methodologies has finally solved both the praziquantel and oxamniquine mechanisms of action; praziquantel activates a transient receptor potential melastatin ion channel [5,6] and oxamniquine acts as a DNA-binding agent after being bio-transformed into its active form via a sulfotransferase [7]. The schistosome drug-discovery community now has actionable information relevant to developing praziquantel analogues with activity against all intra-mammalian lifecycle stages [8] and designing oxamniquine derivatives capable of killing *Schistosoma haematobium* and *Schistosoma japonicum* [9]. Other molecules have been explored for the control of schistosomiasis, such as the broad spectrum anthelmintic amoscanate [10], the dithiolethione derivative oltripaz [11], the benzodiazepine Ro 11-3128 [12] and artemisinin derivatives [13]. Despite their promising anti-schistosomal activity, overuse of the artemisinin derivatives artemether and artesunate for this other therapeutic application would have compromised their effectiveness as a front line antimalarial potentially contributing to the spread of artesunate-resistant *Plasmodium* strains [14]; this particular point has discouraged the further development of these compounds as anti-schistosomals. Furthermore, the antiparasitic activity of the immunosuppressant agent cyclosporine A was discovered by serendipity in the attempt to reduce granuloma formation in a murine model of schistosomiasis [15]. A series of acridine derivatives was subsequently identified as potential anti-schistosomal compounds. These showed activity against all stages of the parasite and were successfully tested in *Cebus* monkeys [16]. However, praziquantel remains the gold standard for treating Schistosomiasis. 

Should praziquantel resistant schistosomes develop (via de novo mutation or standing variation [17]) in an endemic community and outcompete praziquantel sensitive strains (similar to what transpired in Brazil with the rapid emergence of oxamniquine-resistant *S. mansoni* populations), then schistosomiasis control, based on the frontline monotherapy, would be left in a very precarious situation. This concern has fuelled a substantial increase in drug discovery projects over the last 10 years aiming to find praziquantel replacement (or combinatorial) chemotherapies. While some of the more innovative and diverse drug discovery approaches taken by the community have recently been comprehensively reviewed [18], using these to identify and validate new anti-schistosomal drug targets for developing candidate molecules for clinical progression remains challenging [19]. In fact, using some of these approaches [20], only one exceptionally potent, non-toxic, lead compound with characteristics favourable for clinical progression has recently been identified [21]. However, in the same time span, well-curated schistosome genomic/transcriptomic resources for identifying putative parasite vulnerabilities and complementary tools for interrogating gene function have become readily available [22,23]. Therefore, designing approaches to intersect complementary schistosome specific assets with more widely available drug discovery repositories could fast-track the search for new anti-schistosomal interventions. 

With an explicit goal to aid drug discovery, a large-scale RNAi screen of 2216 genes in *S. mansoni* adults was conducted [24]. The results of this screen led to the identification of 261 genes critical to parasite motility/attachment (195) and stem cell maintenance (66); a small subset of these ‘drug targets’ was subsequently validated by whole-organism screens with chemicals (14 in total) predicted from literature searches and information deposited in ChEMBL. ChEMBL is a drug discovery repository that houses medicinal chemistry information derived from research and development processes [25,26,27]. In the context of accelerating schistosome drug discovery, ChEMBL contains extensive details on assays and bioactivities, including compounds/drugs shown (or predicted) to interact with known (mostly human) targets. The stored information is manually curated, well maintained, community accessible and incorporates bioactivity data [28]. Some of this bioactivity data are already derived from *Schistosoma* screens and is included in a sub-repository named ChEMBL-NTD. 

By combining emergent *S. mansoni* functional genomics outputs with growing numbers of chemical matter/target pairings found in ChEMBL, we contend that conditions are now optimal for fast-tracking repositioning campaigns as part of a new era of schistosome drug discovery. Here, we provide further proof of principle data of this concept by identifying several compounds that could be taken forward in the search for urgently-needed anti-schistosomals.

## 2. Materials and Methods

### 2.1. Bioinformatic Pipeline and Compound Selection

The *S. mansoni* protein sequences of 195 genes with severe RNAi phenotypes (‘detachment’ phenotypes) [24] were used to identify compounds predicted to interact with them in the ChEMBL v25 database [28]. This was performed according to [29], with the following modifications. First, for each *S. mansoni* gene, its top BLASTP hit among all ChEMBL targets was identified, as was any ChEMBL targets having BLAST hits with *E*-values less than 10^−10^ of the top hit’s *E*-value. The drugs/compounds with bioactivities against those ChEMBL targets were then extracted. Second, ChEMBL was used to calculate the ‘toxicology target’ interaction component of a compound’s score (probability > 0.5 that the compound interacts with one of 108 toxicology targets previously curated [30,31,32]).

This analysis uncovered 178 compounds (51 phase III/IV approved drugs and 127 medicinal chemistry compounds) potentially targeting 40 *S. mansoni* proteins. Since approved drugs are being studied by many repurposing projects (e.g., as part of the Reframe screening library [33]), 127 medicinal chemistry compounds were selected as representative interactors with these 40 *S. mansoni* proteins (Appendix A). Among these 127 compounds, 14 were excluded from further investigations as they were previously tested *in house* by us in past [24,34] or current studies (unpublished data) (Appendix A, column H). 

The common names (e.g., ‘givinostat’) for the compounds were obtained either from ChEMBL (Appendix A, column F) or by searching PubChem [35] for exact matches to the ChEMBL SMILES string (Appendix A, column G). Although the initial aim was to select medicinal chemistry compounds only, a small number of the 115 compounds have recently been approved as phase III drugs and are listed as such in the latest ChEMBL release (v31): givinostat, pracinostat and abexinostat. 

Quotes from various chemical suppliers (Mcule, Molport and Tocris) were obtained for the remaining 115 compounds and were triaged according to specific requirements (same stereochemistry, minimum available amount of 2–3 mg and purity ≥ 95%). Based on the availability, price and number of *S. mansoni* targets (aiming to cover different target classes), we purchased 80 compounds for testing (Appendix A, column O; Appendix A, column A).

### 2.2. Literature Research

To investigate whether the 80 selected compounds had previously been tested against *S. mansoni*, or other helminths or were similar to other compounds with anti-schistosomal activity, several approaches were used. First, PubMed was searched for specific mentions of the compound common names (Appendix A, columns F, G). To do so, PubMed was queried for the compound common name in the title or abstract and the results were further refined using the string ‘AND (*Schistosoma* or schistosome or schistosomes) to identify *Schistosoma*-specific publications about each compound’ (Appendix A, column I). On the other hand, a string including the compound common name (Appendix A, columns F, G) and key words (such as anthelmintic/helminths/helminth) or various other key genus names (e.g., *Trichuris*, *Brugia* and *Ascaris*.) were used to assess if any of the selected compounds have been previously tested against other helminths (Appendix A, column J). Second, a list of previously published, anti-schistosomal compounds were gathered as described [36]. The SMILES strings were read into OSIRIS DataWarrior (http://www.openmolecules.org/datawarrior/, accessed on 2 February 2019; version 5.2.1, 2021 [37]) and anti-schistosomal compounds similar to our 80 purchased compounds were identified using a ‘Similarity Analysis’ based on the SkelSpheres descriptor. Similarly, DataWarrior was used to compare the structures of the 80 purchased compounds to previously published anthelmintics (against a non-schistosome, helminth species). In addition, we compared the 80 purchased compounds to other compounds named in PubMed abstracts that mention schistosomes or other helminths/nematodes/parasites, as described in [36]. This PubMed-based search is intended to be broader since this focuses not only on the compound common names (similarly to the one described above), but also on their synonyms. Based on these comparisons, we found that, of the 80 purchased compounds, 16 compounds have been previously published (or are similar to a compound published) as having anti-schistosome activity (Appendix A, columns H, I), while an additional 12 compounds (or structurally similar ones) have been previously demonstrated to display additional anthelmintic (e.g., anti-nematode) activities (Appendix A, column J). Of the 80 compounds, 23 were identified to be part of the Reframe screening library [33], by searching https://reframedb.org/ using the InChI key from ChEMBL (Appendix A, column E). 

### 2.3. Target Prediction in S. mansoni and RNA-Seq Meta Data Analysis

The 80 compounds purchased are predicted to target proteins encoded by 29 different *S. mansoni* genes (Appendix A, column K). By searching the literature and curated information on chemical suppliers’ websites, 75 of the 80 compounds have been shown to target homologous proteins in other species (Appendix A, column N). For example, where a particular *S. mansoni* histone deacetylase (HDAC) was predicted as a target based on information in ChEMBL, the compound may also be known to target HDACs in other species.

Insights on the gene expression profile across the *S. mansoni* lifecycle were derived from the RNA-Seq meta database [38]. This is a meta-analysis of published RNA-seq studies on lifecycle stages, including eggs [39], miracidium and sporocyst [40], cercariae and schistosomula [41,42] and male and female juvenile and adult worms [43]. The normalised gene expression values for each gene of interest were obtained by entering the gene identifier (gene ID) into the ‘schisto_xyz’ search engine [44].

### 2.4. Parasite Maintenance and Preparation

The NMRI (Naval Medical Research Institute, Puerto Rican) strain of *S. mansoni* was used to maintain the lifecycle. The intra-mammalian developmental stages were generated by infecting *Mus musculus* (HsdOLa:TO—Tuck Ordinary; Envigo, London, UK); the intra-molluscan developmental stages were propagated through two *Biomphalaria glabrata* strains—the NMRI albino and pigmented outbred strains [45].

*S. mansoni* cercariae were obtained from infected *B. glabrata* snails after 1 h of incubation at 26 °C under intensified lighting conditions. Cercariae were collected in falcon tubes and incubated on ice for at least 1 h prior to transformation. Cercariae were then mechanically transformed into schistosomula as previously described [46].

*S. mansoni* adult worms were recovered by hepatic portal vein perfusion [47] from TO outbred mice (HsdOla:TO, Tuck-Ordinary, Envigo) which were percutaneously infected seven weeks earlier with 180 cercariae/mouse. Following perfusion, adult worms were washed three times in pre-warmed media DMEM (Gibco, Paisley, UK). Following the final wash, adult worm media (DMEM (Gibco) supplemented with 10% (*v*/*v*) FCS (Gibco), 1% (*v*/*v*) L-glutamine (Gibco) and an antibiotic mixture (150 units/mL penicillin and 150 µg/mL streptomycin; Gibco)) was added. All parasites were subsequently transferred into a Petri dish. Before downstream manipulation, any macro residual host material (e.g., mouse hair, blood clots) was removed using a clean paintbrush. Parasites were then incubated in a humidified environment containing 5% CO_2_ at 37 °C for at least one hour, before any experimental use. 

### 2.5. Ex Vivo Schistosomula Screening

Ex vivo schistosomula compound screens were performed as previously described [48,49,50,51]. Briefly, mechanically transformed schistosomula were automatically dispensed into a 384-well plate (120 parasite/well). The parasites were dosed with compounds and incubated in a humidified environment at 5% CO_2_ and 37 °C for 72 h. Following incubation, compound-induced effects were assessed using an in-house facility, Roboworm, which quantifies both larva motility and phenotype [52]. Preliminary compound screens were performed at a single-point concentration of 10 and 50 µM. Three independent screens were performed, including two technical duplicates for each compound/concentration. Each screen contained the positive and negative controls (Auranofin—AUR at 10 μM final concentration in 0.625% DMSO and 0.625% DMSO, respectively). The phenotype and motility scores were used to evaluate whether a compound displayed anti-schistosomula activity; here, −0.15 and −0.35 defined threshold anti-schistosomula values for phenotype and motility scores, respectively. Secondary dose-response titrations were performed for all compounds identified as hits at 10 µM. At least two titrations were performed for each compound. The concentration range included 0.313, 0.625, 1.250, 2.500, 5 and 10 µM (with each concentration point in duplicate). Experimental data (i.e., phenotype and motility scores) were used to compute EC_50_ values using GraphPad Prism 7.02.

### 2.6. Ex Vivo Adult Worm Screening

Adult worms were recovered by hepatic portal vein perfusion and prepared as previously described [34,49,50]. For this specific study, adult worms (1 worm pair/1 mL of adult worm media) were dosed with 10 µM of each compound (in 0.1% DMSO). Negative (0.1% DMSO) and positive control (praziquantel—10 µM in 0.5% DMSO) treatments were included in each replicate (at least three independent experiments). Compound and parasite co-cultures were incubated for 72 h in a humidified environment at 5% CO_2_, 37 °C. Parasite motility after compound treatment was assessed at three time points (24, 48 and 72 h) by a digital image processing-based system (WormassayGP2 [53]) modified after Wormassay [54]. As an additional measurement of parasite health, and since one worm pair was used per assay, notes on the presence of paired or unpaired schistosomes were also collected. Where egg deposition was noticed on the third day, eggs were transferred to a 1.5 mL microfuge tube and centrifuged at 200× *g* for 2 min. With the eggs loosely pelleted at the bottom of the microfuge tube, the remaining media was carefully removed, and the egg pellet was fixed in formalin (10% *v*/*v* formaldehyde). Eggs were counted using a Sedgewick rafter. The final egg count refers to eggs showing normal morphology. Any morphological alterations of eggs (e.g., smaller size, blunt or smaller or total absence of lateral spines, absence of smooth surface) were recorded as well (using similar definitions to other studies [55,56]). 

### 2.7. Statistics

Statistical analysis was performed using GraphPad Prism software (v7.02). For the adult worm screen, a Kruskal–Wallis ANOVA followed by Dunn’s multiple comparisons test was performed to compare each population mean to DMSO mean (the analysis was performed for each timepoint). To statistically evaluate the drug-induced effect on worm fecundity, a Kruskal–Wallis ANOVA followed by Dunn’s multiple comparisons test was performed to compare each egg count mean to DMSO mean. 

### 2.8. Physiochemical Properties Analysis

Compound physiochemical properties in the ChEMBL v25 database were calculated using RDKit (https://www.rdkit.org), as previously described in [28]. The following physiochemical properties for the 80 compounds (Appendix A) presented in this study were downloaded from the web interface: molecular weight, ALogP (calculated value for the lipophilicity), PSA (polar surface area), HBA_Lipinski (count of nitrogen and oxygen atoms in the molecule) and HBD_Lipinski (count of hydrogens attached to nitrogen or oxygen atoms) and Num_RO5_Violations (the number of properties defined in Lipinski’s Rule of 5 that the compound fails). Data analysis was performed with Microsoft Excel version 2010 and finalized by importing the data in OSIRIS DataWarrior (version 5.2.1) [37].

### 2.9. Collection of Literature-Based Cytotoxicity Data from ChemBL

Information about the cytotoxicity profile of each of the 17 hits (Table 1) was extracted from ChEMBL. From this search, only human cell line-derived data was retrieved using the browse search tool available on the web interface of ChEMBL. No relevant data was available for two for the hits (7070707105 and MRT67307). For the remaining compounds, 15 datasets were locally stored as individual .csv files.

Heterogeneous datasets included readouts of cell viability (derived from 3-(4,5-dimethylthiazol-2-yl)-2,5-diphenyl-2H-tetrazolium bromide (MTT) assay, tryphan blue assay, etc.) as well as cell proliferation, growth inhibition or cell death (all measured by propidium iodide/Annexin-V staining based flow cytometry, MTS reduction assay, etc.). Experimental outputs were expressed as CC_50_/IC_50_/EC_50_ or percentage of viable cells compared to the control. Across the datasets, different time points (24, 72 and 96 h) were assessed in these cell assays. 

Extracting the cytotoxicity information from the individual datasets was initially performed in an automatic way using data-mining techniques. First, the ‘assay description’ field of each dataset was scanned for keywords resembling a cell line name based on a set of regular expressions. Next, studies were clustered based on cell lines across the 15 compounds. A colour-coded coverage map (Appendix A) was produced to highlight if there was a common cell line for all compounds and, if not, to identify the cell lines which provided the largest coverage. 

After those data were collected, the corresponding study or publication was manually checked to confirm the experimental data. Cytotoxicity data at 72 h were preferred for comparison across the compounds as this timepoint was used for ex vivo parasitological screening.

## 3. Results

A bioinformatics pipeline (summarised in Figure 1) was built to identify all putative druggable targets within ChEMBL that shared sequence similarity to the 195 *S. mansoni* genes associated with fully-penetrant attachment phenotypes observed after RNAi screening [24]. Initially, all chemical information associated with those targets (i.e., predicted to affect their activity) was filtered based on ad-hoc toxicology, leading to the removal of 17 compounds. Subsequent removal of already approved drugs (51 compounds; these are already found in the Reframe collection and are scheduled to be screened on schistosomes as part of CALIBR’s schistosome drug discovery efforts [33]) resulted in a library of 127 drugs/compounds (Appendix A). We next removed compounds that we have previously tested (highlighted in Appendix A, column H). Based on commercial availability (and purity) at the time this study was conducted, 80 (grey shaded rows in Appendix A) of those 127 drugs/compounds were purchased for our investigation (Figure 1 and Appendix A). As shown in Figure 1, the majority of compounds identified by our approaches for further ex vivo investigations were predicted to target two broad protein classes: kinases (38 out of 80) and epigenetic readers/erasers (22 out of 80).

The ex vivo anti-schistosomal activity of those compounds were initially tested on the larval schistosomula stage using a high content imaging platform [24,34,50] (Figure 2). At the highest concentration tested (50 µM), 37 compounds showed anti-schistosomal activity following 72 h co-incubation (Figure 2a).

At this concentration, we observed a diverse range of phenotypes with several compounds (particularly compound BIO and TAE684, but also Radicicol and Trichostatin A) showing an auranofin-like effect (i.e., an elongated shape and increased granularity). Other compounds (e.g., the kinase inhibitors AZ 191, Tozasertib, KW2449, as well as the HSP inhibitor NVP_BEP800) induced generalised granularity and a more rounded effect on the schistosomula (Figure 3).

Only 17 of these compounds retained their anti-schistosomal activity at 10 µM (Figure 2b). Not surprisingly, most of the hit compounds (10/17 at 10 µM) were predicted to interact with kinases due to the higher proportion of kinase modulators included in the initial library.

The ex vivo anti-schistosomula potencies of these 17 hits (Figure 4) were next explored during dose-response titrations (Figure 5). 

While some compounds predictably lost their anti-schistosomula activity at lower concentrations (e.g., AZ 191, BIO, G 5555, MRT67307), some remained potent even at low micromolar/high nanomolar concentrations (e.g., Staurosporine, Epoxomicin, Luminespib). Except for minimal experimental variations expected during biological replications, all compounds differentially reduced both anti-schistosomula phenotypic metrics measured (i.e., phenotype and motility, Figure 5a,b). The HSP inhibitor Luminespib and the non-selective protein kinase inhibitor Staurosporine demonstrated the greatest anti-schistosomula activities (i.e., the lowest EC_50_ values) (Table 1).

The 17 anti-schistosomula hit compounds (at 10 µM) were next assessed for ex vivo activity on adult worm pairs using phenotypic quantification of motility and pairing at 24, 48 and 72 h (Figure 6). 

Based on the number of compounds assessed, a single concentration (10 µM) was evaluated. Here, three compounds (SNX 2112, luminespib and trichostatin A) showed little or no activity on adults (when compared to DMSO controls) at 72 h. Compound 7070707105 reduced worm motility within 24 h; however, this was only a transient effect as worm movement began to recover by 72 h. A more time dependent effect was observed for five compounds (radicicol, NVP-BEP800, AZ 191, MRT67307 and A-674563). We also identified four fast-acting compounds (epoxomicin, CGP 60474, PHA665752 and staurosporine) with total inhibition of worm movement within the first 24 h of treatment. 

While most of the compounds (12; white boxes on top of Figure 6) also inhibited schistosome pairing during co-culture, five compounds (fimepinostat, trichostatin A, SNX 2112, luminespib and 7070707105) did not (blue boxes). As previously demonstrated [57,58], adult male-female pairing is necessary for female sexual maturation and egg production. Therefore, we next asked if compound-induced inhibition of pairing was associated with egg production deficiencies (Figure 7). 

For the majority of the compounds under investigation, there was a statistically significant reduction of in vitro egg production (compared to the DMSO control) (Figure 7a). Total inhibition of in vitro laid egg (IVLE) production was positively correlated with those 12 compounds that also inhibited adult worm pairing. For five compounds that did not affect schistosome pairing (fimepinostat, trichostatin A, luminespib, SNX 2112 and 7070707105, Figure 6), there were more eggs produced during ex vivo co-culture (but still fewer than DMSO controls). Interestingly, compounds SNX 2112 and 7070707105 also affected egg morphology (Figure 7b). In addition to quantifying adult worm motility and egg production defects, we qualified compound-specific phenotypes by bright field microscopy (Appendix A). These phenotypes were defined by two main signatures: bubble formation on the adult worm surface (e.g., AZ 191, radicicol and staurosporine; arrows) and wide-scale tegument sloughing/degeneration (e.g., BIO, TAE684, MRT67307, PHA665752, arrows). 

To provide further context on the drug-like properties of each compound tested, we interrogated the extensive portfolio of information available in ChEMBL. First, we focused on the Lipinski’s rule of five to predict the oral bioavailability of those compounds: molecular weight MW < 500, calculated octanol/water partition coefficient clogP < 5, number of hydrogen bond donor atoms HBD < 5 and number of hydrogen bond acceptor HBA < 10 [59]. Those parameters are conveniently summarised in the compound card of ChEMBL (summarised in Appendix A). Looking at each parameter individually, we noticed that most studied compounds fell within the literature threshold of acceptability (Figure 8). 

Of the 37 compounds active at 50 µM, only 9 compounds had one or two violations to the rule of five (Appendix A and Figure 8a–c). Only 5 of the 17 most active compounds also contained one or two violations to these rules (Appendix A), which is not surprising as many orally bioavailable, drugs as well as >180K ChEMBL 24.1 entries, exhibit Lipinski constraints [60]. 

We secondly investigated the polar surface area (PSA) of these 80 compounds (Figure 8d). This parameter has been used successfully to predict the absorption of drugs [61] with values below 140 Å being recommended for good oral absorption [62]. Based on this parameter, we observed that only 9 out of 80 compounds exceeded this limit (Appendix A). 

The compounds presented in this study have been differentially progressed in the drug discovery pipeline for multiple applications. In fact, three compounds have recently been approved as phase III drugs (e.g., givinostat, pracinostat and abexinostat). Another nine compounds are in phase II development and eight compounds are currently in phase I development (Appendix A).

Evaluation of the safety profile of a compound is an essential step in the anti-schistosomal drug discovery pipeline in order to prioritise chemical entities that are not overtly cytotoxic. For this reason, the cytotoxicity/antiproliferative data of 15 out of 17 compounds listed in Table 1 were obtained from ChEMBL. Unfortunately, no relevant data was available for 7070707105 and MRT67307. 

Analysis of the ChEMBL data highlighted approximately 900 different cell lines which have been previously subjected to cell cytotoxicity assays with our 15 anti-schistosomal hits; however, we could not identify a single cell line uniformly used for all 15 compounds (Appendix A). We, therefore, selected the five cell lines providing the highest coverage: K562 (chronic myelogenous leukemia cell line), HCT116 (colon cancer cell line), A549 (human lung adenocarcinoma cell line), MCF7 (epithelial cell line) and Huh-7 (human hepatocellular carcinoma cell line) (Table 2). We observed that compounds like staurosporine and trichostatin A have been extensively explored on different cell line panels with results derived from several studies demonstrating a high degree of reproducibility. The majority of the 15 anti-schistosomal hits had a cytotoxicity profile in the range of medium-low nanomolar across the five selected cell lines. As the most active compound on the schistosomula stage presented in this study with an EC_50_ in the range of high nanomolar (EC_50_ = 0.194 µM for luminespib, Table 1), we conclude from these comparisons that the host/parasite selectivity ratio is quite narrow. However, medicinal chemistry optimisation could improve both ex vivo anti-parasite activity and host cytotoxicity. Furthermore, the compound concentration showing efficacy in vivo might still be quite different from the concentrations used in ex vivo settings. One outlier to this generalisation is the kinase inhibitor BIO, which displays a better cytotoxicity profile (Table 2). As this compound showed a moderate ex vivo anti-schistosomal activity against schistosomula and a potent fast-acting activity against adults, BIO represents a good lead candidate with its scaffold to be further explored for future anti-schistosomal applications. 

We subsequently focused our attention on the three kinase inhibitors (CGP 60474, TAE684 and PHA665752), which had the most severe effect on both adult worm motility and fecundity (Figure 6 and Figure 7) as well as sub-micromolar (e.g., CGP 60474) and low micromolar (e.g., TAE684 and PHA665752) activities (EC_50_ values) on schistosomula (Table 1). While no cytotoxicity data were available for these three small molecules on the five selected cell lines (Table 2), further investigation of ChEMBL highlighted that those compounds were all part of the Genomics of Drug Sensitivity in Cancer project (available from “https://www.cancerrxgene.org/”, accessed on 01 April 2022) [63]. As shown in Table 3, the three compounds demonstrated variable cytotoxicity depending on the specific cancer cell line tested. However, based on these data, TAE684 and CGP 60474 showed good toxicity profiles. For this reason, these two anti-schistosomal compounds could be prioritised for further investigations. 

## 4. Discussion

The use of ChEMBL to aid the translation of genomic information into new anthelmintic drugs was first conceptualised in 2018 [64], applied to the comparative genomics analysis of 81 nematode and platyhelminth worms in 2019 [29] and then leveraged for a subset of *S. mansoni* genes in 2020 [24]. The present study represents a comprehensive follow-up project showcasing how to effectively couple drug discovery repositories (e.g., ChEMBL) and *S. mansoni* specific resources (e.g., whole organism ex vivo assays, schistosome functional genomics resources—RNAi screens) to rapidly identify target/compound combinations that could enter the drug discovery pipeline. 

The 17 compounds with at least 10 µM activity against schistosomula included four different HSP90 inhibitors: SNX 2112, radicicol, NVP-BEP800 and luminespib (Figure 2c, Table 1). There are various mechanisms of HSP90 inhibition [65], but these four all act by binding to the N-terminal ATP domain of HSP90. Known HSP90 inhibitors have been categorised into five chemical classes: (i) natural products and their derivatives, (ii) purine-based, (iii) benzamide, (iv) resorcinol-containing and (v) miscellaneous [66]. The resorcinol-derived radicicol (Appendix A) is a known HSP90 inhibitor [67], which was previously shown to be active against *Schistosoma japonicum* [68]. In this study, we showed a moderate activity of radicicol on *S. mansoni* schistosomula (EC_50_ = 1.828 µM), a time-dependent effect on adult worm motility (Figure 6), an egg production deficiency (Figure 7) and tegumental damage (with bubble-like structures observed on the tegument as shown in Appendix A). Interestingly, the related compound zearalenone did not show any activity on *S. mansoni* schistosomula (as this was not active at 50 µM, Appendix A) and, therefore, was not progressed on adult worms. Both compounds are members of the resorcylic acid lactone class; however, the reactivity of the radicicol epoxy group might be the main driver of the higher activity of radicicol compared to its structural analogs. Two isoxazole resorcinol HSP inhibitors (luminespib and NMS-E973) were tested, but only luminespib (known also for its activity against filarial nematodes [69]) was active against schistosomula (EC_50_ = 0.197 and 0.191 µM, for phenotype and motility, respectively), although it showed little activity against adults (Figure 6). Second to luminespib, NVP-BEP800 was the second most anti-schistosomula active of the HSP90 inhibitors analysed here (EC_50_ = 0.546 µM), also showing potent and time dependent effects on adult worm motility (Figure 6) and IVLE production (Figure 7). NVP-BEP800 belongs to the purine-like class of HSP inhibitor and is a structurally related compound of BIIB021, which was previously shown to have activity against *S. mansoni* [24]. NVP-BEP800 also shows a small degree of similarity to the purine-based HSP90 inhibitor PU H71, which was found to be active against *S. mansoni* larvae at 50 μM (Figure 3). BIIB021, PU H71 and NVP-BEP800 are all advanced HSP inhibitors with the first two compounds containing the purine scaffold that mimics the shape adopted by ATP when bound to HSP90. This scaffold is replaced by a thienopyrimidine in NVP-BEP800. SNX 2112 was the only anti-schistosomula active HSP inhibitor (Figure 3) that carried a benzamide scaffold, but it showed little activity against adult worms (Figure 5 and Figure 6). These results collectively suggest that the purine-like and the resorcinol-based class HSP inhibitors are interesting anti-schistosomal hits, which could be explored further. From our ChEMBL-based pipeline (Appendix A), we predicted the target of these HSP90 inhibitors in *S. mansoni* to be Smp_072330 (Appendix A, column K), which has been previously identified as *S. mansoni* and *S. japonicum* HSP90 [70]. Those HSP inhibitors showed a better activity profile against schistosomula, rather than the adult stage. However, this could be partially explained by higher levels of *smp_072330* expression in schistosomula compared to adult worms (Appendix A). 

Our library of compounds included only two proteasome inhibitors, the tripeptide epoxyketone epoxomicin and the peptide aldehyde inhibitor MG-132 (Appendix A). Epoxomicin is among the most potent of the 17 active compounds against schistosomula (average EC_50_ = 0.226 µM), was fast-acting on adults (Figure 6) and significantly inhibited IVLE production (Figure 7). On the other hand, MG-132 was only active at 50 µM on schistosomula (Appendix A). Previous evidence showed that 50 μM of MG-132 caused peeling, outbreaks and swelling in the tubercles of the adult worm tegument, but these phenotypes did not lead to parasite death [71,72]. At 50 µM, MG-132 was active against schistosomula in our investigations (Appendix A), but at lower concentrations (10 µM) was inactive. Because of this, MG-132 was not triaged for testing against adult worms in our studies. Biochemical evidence has confirmed differential activity of epoxomicin and MG-132 on the ubiquitin–proteasome proteolytic pathway in parasite crude extracts, particularly interacting with the 20S proteasomes from *S. mansoni* adult worms and cercariae [71]. The proteasome is a quite attractive area of investigation, with other related tripeptide/tetrapeptide epoxyketone proteasome inhibitors (carfilzomib and carmaphycin B [73] as well as ixazomib, NMS-873 and CB-5083 [24]) previously shown to have activity against *S. mansoni*. In fact, compounds such as carfilzomib and carmaphycin both have an epoxyketone group, similarly to epoxomicin. Clearly, our results indicate that epoxomicin and MG-132 have a different ex vivo activity profile despite their structural similarity. However, the lack of MG-132 potency in our assays may be due to: the aldehyde group being oxidized rapidly within the worm [74], the poor penetration of the compound or the rapid dissociation half-life of peptide-aldehydes from the proteasome, compared to the irreversible peptide epoxyketone (epoxomicin) [73]. Most proteasome inhibitors, including MG-132 and epoxomicin, target the β5 subunit of the proteasome [73]. From our ChEMBL-based pipeline, we predicted the target of epoxomicin and MG-132 in *S. mansoni* to be Smp_212880, which based on WormBase ParaSite (version WBPS16) is the many-to-many ortholog of human alpha and beta subunit genes PSMB5, PSMB6, PSMB8, PSMB9, PSMB10, PSMB11, PSMA7 and PSMA8. The expression profile of this gene (*smp_212880*) shows a peak in the sporocyst stage, followed by a constant level in all intra-mammalian stages of the parasite (Appendix A). 

Another interesting target identified in our pipeline is the histone deacetylase (HDAC) class with 16 putative HDAC inhibitors being selected for ex vivo testing (Appendix A). Among those, the natural product Trichostatin A (TSA) was one of the first generation of HDAC inhibitors to be identified [75]. It had been extensively explored and proven to block metamorphosis of *S. mansoni* free-swimming miracidia into the intra-molluscan sporocyst [76] as well as cause the mortality of schistosomula at 1 and 2 µM over a 7-day co-culture (with media exchange occurring daily) [77]. Here, we confirm the anti-schistosomula effect of TSA at 10 and 50 µM following 72 h continuous co-incubation. While our anti-schistosomula activities did not translate to adults (at 10 µM for 72 h continuous co-culture, Figure 6), the mechanism of TSA action in schistosomula likely involves histone H4 hyperacetylation by inhibition of global HDAC activity and apoptosis [77]. 

TSA contains a specific structural feature, hydroxamic acid, which was identified as a metal-binding moiety that coordinates the cation Zn^2+^ within the HDAC active site. Therefore, thousands of synthetic HDAC inhibitors containing this motif have been reported [78]. Among those, there are three additional compounds (abexinostat, MPT0E014 and fimepinostat), presented herein, with this HDAC inhibitor-specific feature (i.e., the hydroxamic acid). However, these compounds showed activity against *S. mansoni* schistosomula only at 50 μM. Only fimepinostat had an EC_50_ below 10 µM. However, this compound appears to have a promiscuous mode of action, being a HDAC as well as a phosphoinositide 3-kinase (PI3K) inhibitor [79]. Furthermore, it showed little activity against adult worms (Figure 6 and Figure 7).

Other first generation HDAC inhibitors (e.g., vorinostat, belinostat, panobinostat and romidepsin) were previously investigated on *S. mansoni* [80,81] and most failed to show any activity against schistosomula, except for panobinostat and romidepsin. These two compounds additionally showed a modest or complete inhibition of adult worm pairing and egg production at 10 µM concentration [81]. Of note, panobinostat was also investigated by Wang et al. [24] against both *S. mansoni* schistosomula and adult worms, confirming the moderate anti-schistosomal activity. Our study has further explored the chemical space around these HDAC inhibitors since, in our hands, the heterocycle derivative of vorinostat (pyroxamide, Appendix A) did not show any activity against *S. mansoni* (Appendix A). Similarly to panobinostat, the close analogue dacinostat (Appendix A) was only active on schistosomula at the higher concentration tested (50 µM) and, therefore, not progressed in adult worms. A common scaffold is present in both dacinostat and panobinostat, which could be further explored and optimised for further anti-schistosomal drug development. 

The use of ortho-aminoanilides (benzamides, Appendix A) as a replacement of the hydroxamic acid moiety, seems to improve HDAC inhibition and prevent the mutagenicity issues associated with hydroxamate function [82]. In this study, however, neither of the two representatives of this class (BRD-6929 and BRD4884) showed significant activity on schistosomes (Appendix A). 

In previous investigations, hinokitiol, a monoterpenoid HDAC inhibitor [83], caused ultrastructural changes to the surface and parenchyma of *S. mansoni* cercariae [84]. However, our studies of this putative HDAC inhibitor on schistosomula did not demonstrate activity. This could be partially due to different lifecycle stages examined (schistosomula here rather than cercariae in [84]) or concentrations used (50 or 10 µM here compared to 50 µg/mL-approximately 304 µM in [84]).

As demonstrated in the narrative above, targeting the *S. mansoni* histone deacetylation pathway has been investigated as an approach to treat schistosomiasis [85]. Despite the number of hydroxamic acid HDAC inhibitors (HDACi) that have reached clinical development, concerns with pharmacokinetic properties (such as rapid metabolism, off-target effects due to non-selective metal binding and relatively poor bioavailability) were identified [75]. These properties might reduce the translation of promising ex vivo anti-schistosomal activities to an in vivo animal model. Moreover, the development of schistosome-specific HDACi still remains a challenge due to the high sequence homology between the human (HDAC1/2) and schistosome (e.g., Smp_005210, annotated as smHDAC1 [86]) targets (Appendix A). The predicted target (*smp_005210*) of these HDACi is highly expressed in the miracidia and sporocyst stage compared to schistosomula (Appendix A), which likely explains the dramatic effect of TSA on miracidia-sporocyst transformation.

By far the largest group of compounds identified by our approach and tested on schistosomes were putative protein kinase (PK) inhibitors (38 out of 80 compounds). Kinases are promising molecular targets for drug development due to their contributory role in cancer and autoimmune disorders with more than 72 FDA-approved drugs currently in use [87]. Drug repurposing of kinase inhibitors represents a strategy for schistosomiasis drug discovery, as recently summarised in [88]. *S. mansoni* protein kinases have also recently been explored by computer-aided drug design (CADD) to filter a large library (the Managed Chemical Compound Collection (MCCC) of approximately 85,000 unique chemical compounds) aiming to identify novel starting points for schistosome drug discovery [89]. 

In our study, four main chemical subclades can be identified among the 20 putative PK inhibitors showing anti-schistosomal activity at 50 µM (Appendix A): the phenylamino-pyrimidine-like-, the indolone and related-, the amphetamine- and pyridonopyrimidine- classes (Appendix A). In previous investigations, tyrosine kinase inhibitors containing a phenylamino-pyrimidine (PAP) moiety, similar to imatinib, have demonstrated activity against *S. mansoni* and other flatworms [88]. Here, we add to this information and show anti-schistosomula activities of other PAP-like compounds, such as CGP 60474 (inhibitor of cyclin-dependent kinase (CDK) and protein kinase C (PKC)), MRT67307 (inhibitor of IKK-ε and TBK1 and UKL1 and ULK2 [90]), 7070707105 (inhibitor of ALK5 and PKN3 kinase), AZ 191 (inhibitor of DYRK1B) and TAE684 (another inhibitor of ALK). While compound 7070707105 did not affect worm pairing, it did transiently affect adult worm motility at 24 h; full recovery of motility and the production of abnormal eggs was observed at 72 h (Figure 6 and Figure 7). Compound 7070707105 is structurally related to vandetanib, a PK inhibitor identified during a repurposing study of anticancer PKs from the Developmental Therapeutics Program (DTP) of the National Cancer Institute, USA, against larval and adult stages of *S. mansoni* [91]. In that study, vandetanib induced a 48.1% worm burden reduction following a single oral dose of 400 mg/kg body weight. Therefore, pursuing compound 7070707105’s in vivo efficacy in an animal model of schistosomiasis seems warranted.

AZ 191 and MRT67307 were not the most potent anti-schistosomula PAP-like kinase inhibitors (Table 1, EC_50_ = 3.402 and 3.132 µM, respectively) nor the most fast-acting compounds on adults (no significant inhibition of worm movement at 24 h); however, they substantially affected worm motility by 72 h. Those compounds also prevented pairing (Figure 6), inhibited IVLE production (Figure 7) and induced surface damage (Appendix A). Similar to AZ 191, AZ Dyrk1B 33 is another inhibitor of the dual-specificity tyrosine phosphorylation-regulated kinase (DYRK [92]), but is active only at 50 µM on schistosomula. As part of a library of 2040 vertebrate kinase inhibitors, a scaffold similar to AZ 191 (AZD-3463) was previously investigated on the model nematode *Caenorhabditis elegans* aiming to develop new anthelmintics [93]. In contrast to these modestly acting PAP-like kinase inhibitors, CGP 60474 demonstrated potent anti-schistosomula and adult worm activities (Table 1, EC_50_ = 0.409 µM; Figure 6) as well as anti-fecundity effects (Figure 7). The final PAP-like kinase inhibitor assessed here, TAE684, which was moderately active against schistosomula (Table 1, EC_50_ = 2.399 µM), caused a substantial time-dependent effect on adult worm motility after 48 h and led to egg production inhibition as well as tegument degeneration (Figure 6, Figure 7, Appendix A). The remaining two pyrimidine-based compounds (AST-487 and tozasertib or VK-680) are good dual leucine zipper kinase (DLK) binders but most commonly used as pan-kinase inhibitors due to their poor selectivity profile [94]. These two compounds showed anti-schistosomal activity only at the higher concentrations tested (50 µM). The activity differences between these final two compounds compared to other members of the phenylamino-pyrimidine-like subclade could be due their selectivity profile or the replacement of the linker in the phenylamino-pyrimidine group (Appendix A).

Staurosporine is an alkaloid known to have anti-fungal activity likely due to its non-selective inhibition of Ca^2+^/calmodulin-dependent PK II (CaMKII, Appendix A). This compound was also one of the most potent anti-schistosomula compounds tested (Table 1, EC_50_ = 0.202 µM). It had a potent fast-acting effect on adult worms and induced substantial tegumental damage (Figure 6, Figure 7, Appendix A). The anti-schistosomal activity of staurosporine has previously been reported [95,96]. These investigations showed a promising effect of this compound used in synergy with PZQ [95]. However, those studies also highlighted the non-selectivity and increased toxicity of this compound, which required further investigation. However, we have included staurosporine in the current study to test it in house and use this as reference for the other two indolone-containing compounds (BIO and PHA665752) selected by our bioinformatic pipeline. BIO and PHA665752 showed moderate in vitro anti-schistosomal activity against schistosomula (EC_50_ = 2.340 and 4.513 µM, respectively) when compared to staurosporine. However, similar to staurosporine, both PHA665752 and BIO showed potent fast-acting activity against adults (Figure 6), completely inhibited egg production (Figure 7) and induced tegument damage (Appendix A). GF 109203X was another derivative of this class. Being active only at the higher concentration tested (50 µM) on schistosomula (Appendix A), it was not further progressed in adult worms. The previous description of this compound’s activity on adult *S. mansoni* worms only at high concentrations (20–50 µM) [97] supports our findings. The pyridonopyrimidine class G 5555 exemplar, an inhibitor of p21-activated kinase 1 (PAK1 [98]), demonstrated moderate activity on schistosomula (EC_50_ = 2.836 µM) and quick effects on adult worms (Figure 6) that led to egg production inhibition (Figure 7). In contrast, the related compound PD173955 did not show any activity in our phenotypic screens. Our study also identified the amphetamine class A-674563 exemplar (inhibitor of Akt1 and PKA [99]), which does not show any relevant similarity to previously published anti-schistosomal or more broadly acting anthelmintic compounds, but had a moderate effect on schistosomula (EC_50_ = 2.928 µM), a time-dependent effect on adult worm motility (Figure 6) and inhibited egg production (Figure 7).

A careful analysis of the literature identified a few kinase inhibitors, previously reported as having anti-schistosomal activity, but showing little or no activity in our study. Firstly, SB203580, an inhibitor of p38 MAPK (Appendix A), was shown to affect the transitioning of miracidia into sporocysts [100] as well as the swim velocity of miracidia by interaction with putative *S. mansoni* p38 MAPK [101]. However, no experimental evidence was provided by the authors of the study to suggest that SB203580 had any activity on either schistosomula or adult schistosomes. The low expression of the putative target for SB203580 (Smp_047190, Appendix A) in the lifecycle stages investigated here may also provide an explanation for the absence of in vitro activity. OR7811 (or LY-364947, Appendix A), an inhibitor of type I TGFβ receptor (TβRI), was shown to reduce *S. mansoni* mitogenic activity and egg production (at 300 nM, [102]), as well as inhibit SmTβRI kinase activity in vitro (derived from heterologous expression of SmTβRI’s intracellular domain in *Xenopus laevis* oocytes [103]). In our hands, SB 431542 (but not OR7811) was only active on schistosomula at 50 µM. It is not surprising that different conclusions are being drawn when the same kinase inhibitors are being used to study distinct aspects of schistosome biology. 

Drug-like molecules are classically defined by the Lipinski’s rule of five [59,87]. Failure to comply to this physicochemical profile might translate to poor pharmacokinetic properties. As shown in Figure 8 and Appendix A, 12 of the 17 schistosomula hits presented in this study satisfy these requirements. This represents a promising feature for the translation of the ex vivo activity of these compounds into a murine model of schistosomiasis. Nevertheless, the number of orally bioavailable drugs and drug candidates with one or more violations to the Lipinski rule is constantly increasing, so the other five hits (fimepinostat, epoxomicin, MRT67307, PHA665752 and TAE684, Appendix A) could be revisited at a later stage despite having one or two violations to the Lipinski rule. 

As mentioned in the methodology, we have deliberately removed any approved drugs in our initial compound selection. Even though few compounds are either in phase I, II or III clinical trials (Appendix A), the majority have not reached those advanced phases of drug development, so further investigation will be required. However, the detailed portfolio of data available in ChEMBL can still help with triaging the compounds in the schistosomal drug discovery pipeline. In fact, as described in this study, we extrapolated, from ChEMBL, the cytotoxicity data for each of the 17 hits. To further improve this ChEMBL-based workflow, we are planning to evaluate the cytotoxicity data at an early stage of the compound selection and eventually use this as a filtering criterion aiming to address the host safety profile at a very early stage. Having said that, this improvement strongly relies on data availability, which could be more expansive for compounds in phase III clinical trials compared to other less advanced compounds.

## 5. Conclusions

By combining emergent *S. mansoni* functional genomics outputs with the increased chemical matter/target pairings found in ChEMBL, we contend that conditions are now optimal for fast-tracking whole proteome (large-scale) or target-based (small-scale) repositioning campaigns as part of a new era of schistosome drug discovery. Keeping in account that schistosome researchers now have access to a complement of well-curated genomic, transcriptomic and rapidly-expanding functional genomics information, this approach could be further explored and implemented in the future. In fact, wider approaches looking at the entire parasite proteome as well as more stringent criteria (target expression profile) are some of the improvements we envisage which, will generate less, but more likely progressable, drug candidates for further analyses. Therefore, the implementation of the current pipeline with information regarding temporal and spatial gene expression of schistosome targets could represent a substantial improvement in short-listing druggable candidates.

## Figures and Tables

**Figure 1 pharmaceutics-15-01359-f001:**
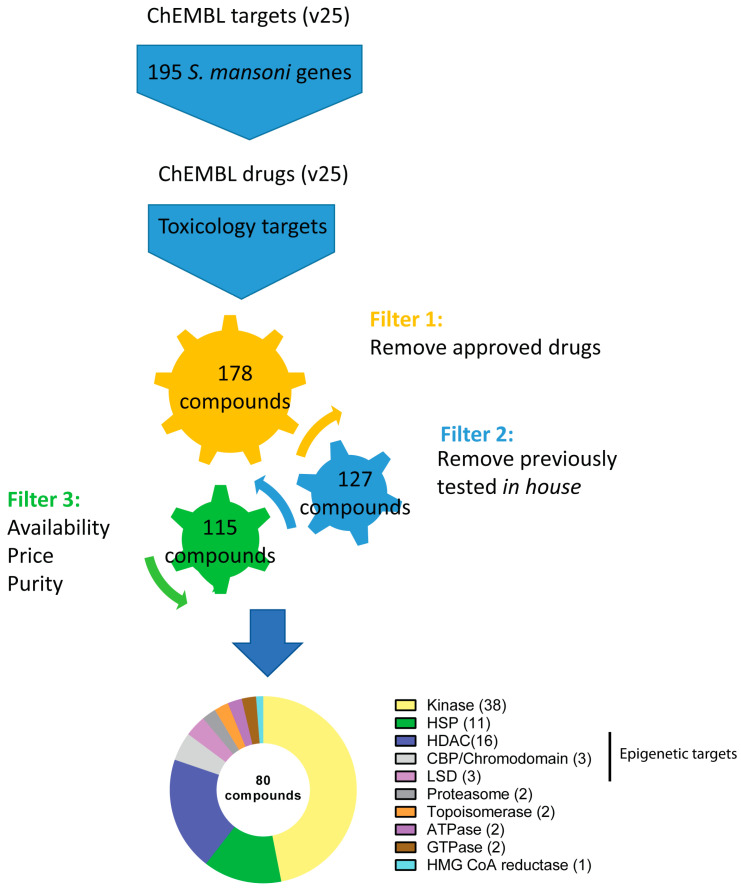
Schematic representation of the bioinformatics pipeline used to identify putative anti-schistosomal compounds. This representation summarises the inputs, outputs and filter criteria used in this study, leading to the selection of 80 compounds for ex vivo parasite screening (see methodology for full details). The pie chart represents the target class distribution: ten different target types are listed and the number of compounds for each target are shown in brackets.

**Figure 2 pharmaceutics-15-01359-f002:**
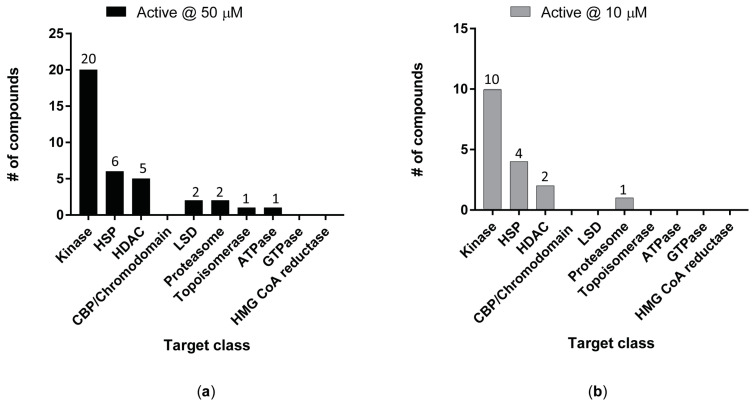
Ex vivo schistosomula hit distribution across the different target classes. Following single-point concentration screens at 50 µM (**a**) and 10 µM (**b**), the number of active compounds in each particular target class is shown.

**Figure 3 pharmaceutics-15-01359-f003:**
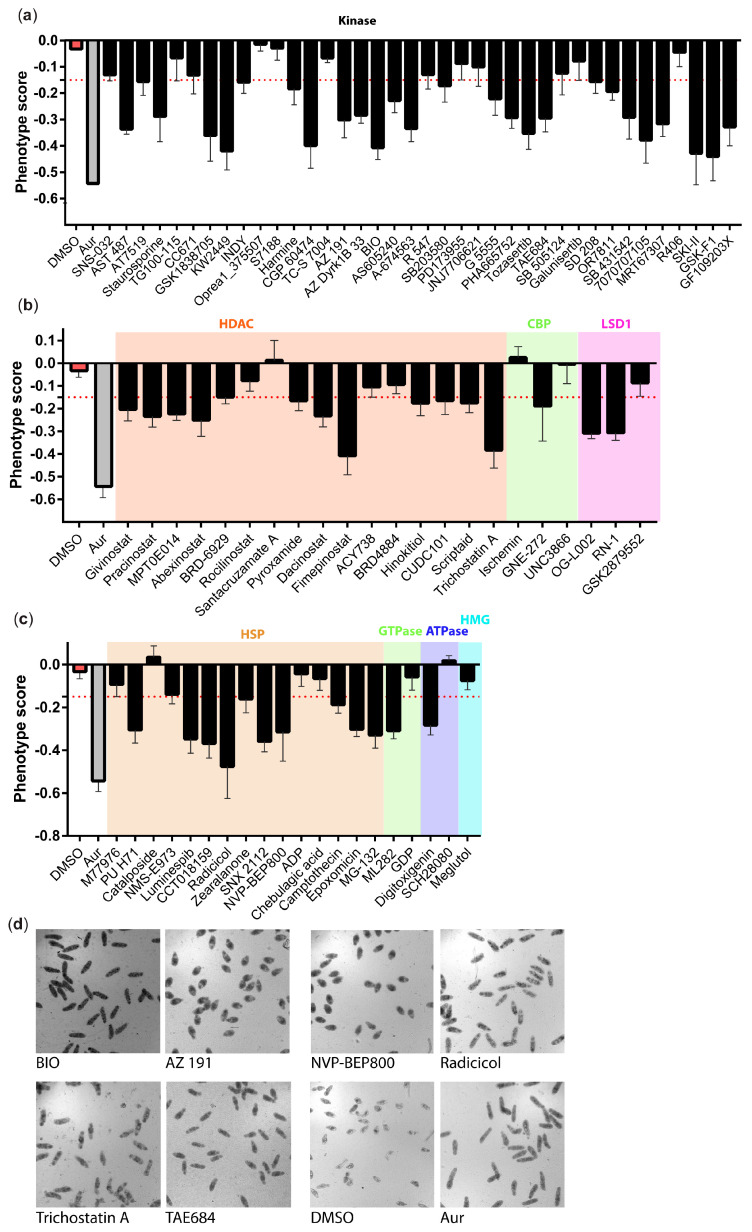
Ex vivo anti-schistosomula effects of the 80 compounds. Phenotypic scores for all 80 compounds tested at 50 µM are illustrated as bar charts: (**a**) represents the kinase inhibitors, (**b**) represents the epigenetic inhibitors and (**c**) represents the remaining compounds. Each bar chart shows the mean phenotypic score (across three independent replicates with two technical replicates each) along with the standard deviation. The negative and positive controls (DMSO and Auranofin—AUR) are included in each panel. The red dotted line indicates the threshold value of −0.15; all compounds with a phenotype score lower than the cut-off value are defined as hits. (**d**) Bright field images of representative compound-induced phenotypes observed following 72 h co-incubation. The images were acquired using an ImageXpressXL high content imager (Molecular Devices, Wokingham, UK) with a 10x objective.

**Figure 4 pharmaceutics-15-01359-f004:**
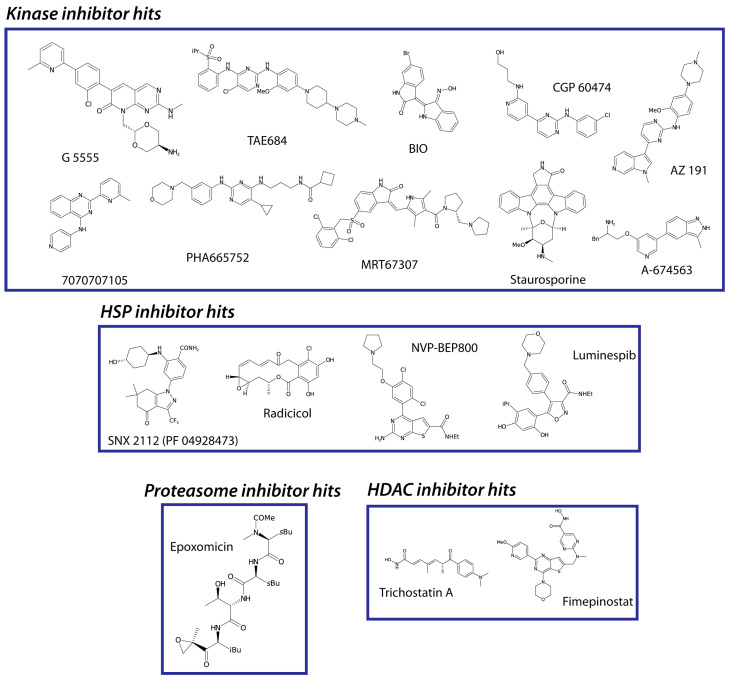
Structures of the anti-schistosomula compounds. The chemical structures of the 17 hits (at 10 µM) are shown here and are grouped by target class. Compounds structures were visualised using the open-source chemistry development kit (CDK), Depict v1.9.22, CDK v2.7.1. Abbreviations: Et = ethyl group; Me = methyl group; iBu = isobutyl; sBu = sec-butyl; Bn = benzyl.

**Figure 5 pharmaceutics-15-01359-f005:**
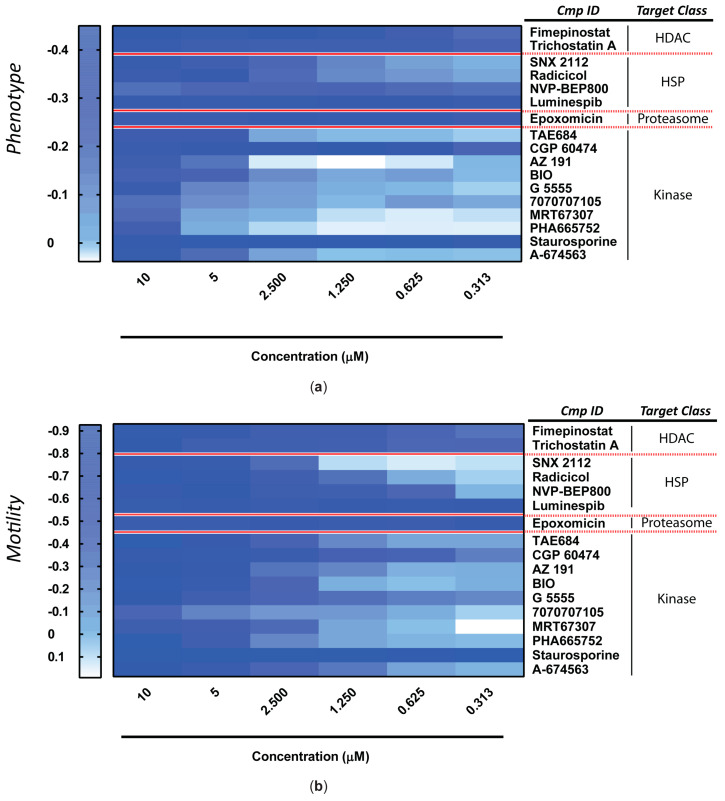
Dose response titrations of the anti-schistosomula hits. Titrations were performed for each of the 17 hit compounds (at 10 µM) identified in the preliminary screens. Each compound concentration (0.313, 0.625, 1.250, 2.500, 5 and 10 µM) was evaluated in duplicate. The mean phenotype (**a**) and motility (**b**) scores (from three independent experiments, each containing two technical replicates for each concentration point) are represented as heat maps.

**Figure 6 pharmaceutics-15-01359-f006:**
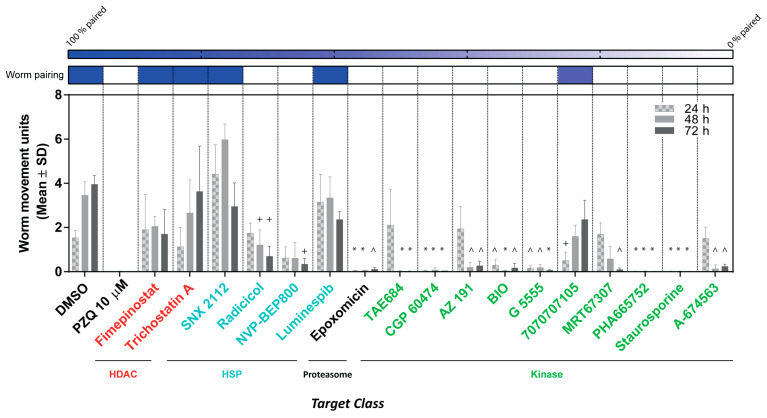
Assessment of the anti-schistosomula hits on adult worms. Each of the 17 compounds was tested at 10 µM on a single adult worm pair. Worm movements were recorded at 24, 48 and 72 h. The mean worm movement across three independent replicates is presented as a bar chart. Following three days of co-incubation; worm pairing was also recorded and here shown as a heat map (on top of the bar chart). A Kruskal–Wallis ANOVA followed by Dunn’s multiple comparisons test was performed to compare each population mean to DMSO mean (the analysis was performed for each timepoint). For both panels, *, ^ and + represent *p* < 0.0001, *p* < 0.0021 and *p* < 0.0332, respectively.

**Figure 7 pharmaceutics-15-01359-f007:**
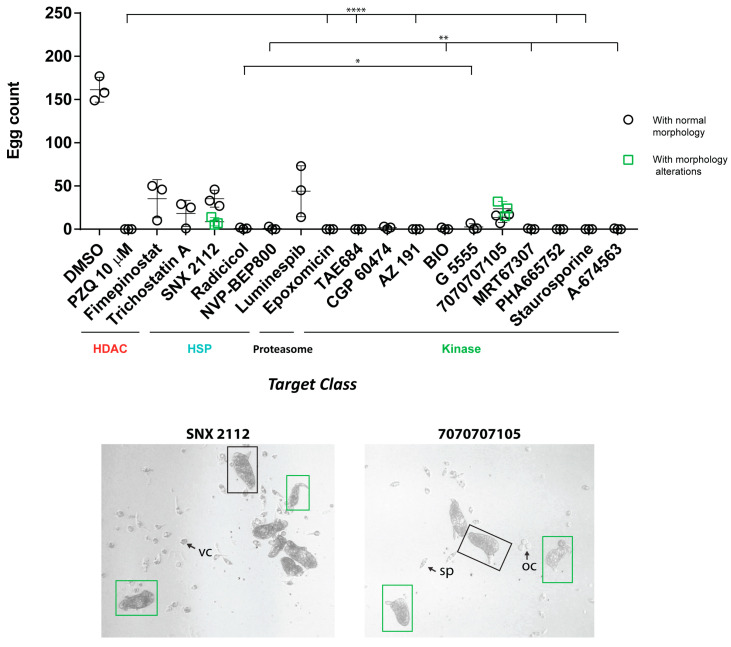
Adult worm anti-fecundity effects of the anti-schistosomula hits. Following 72 h incubation with each of the 17 compounds, in vitro laid eggs were enumerated. For each compound, each data point corresponds to the egg count of an independent screen (three replicates in total). A Kruskal–Wallis ANOVA followed by Dunn’s multiple comparisons test was performed to compare each egg count mean to DMSO mean (with *, ** and **** representing *p* < 0.0332, *p* < 0.0021 and *p* < 0.0001, respectively). Morphologically normal eggs with lateral spine were enumerated (black boxes). For two compounds (SNX 2112 and 7070707105), eggs with morphological alterations (green boxes) were observed (brightfield images are shown here for each compound). Cellular material was also observed in the well (oocytes (oc), spermatozoa (sp) and vitelline cells (vc) are indicated with arrows).

**Figure 8 pharmaceutics-15-01359-f008:**
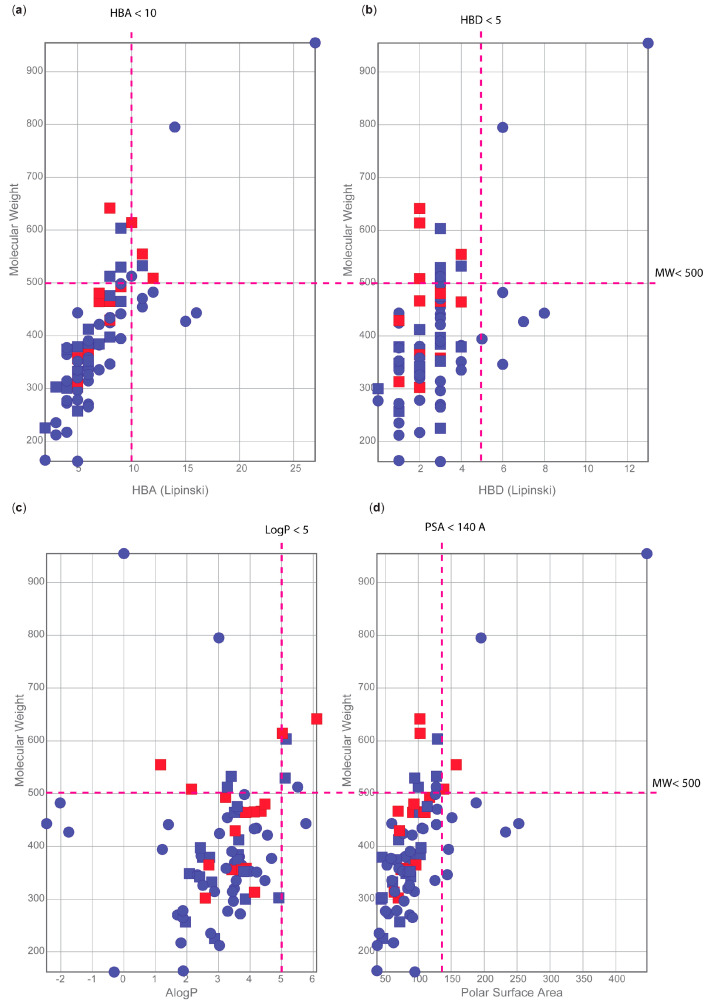
Analysis of the physicochemical properties of the 80 compounds initially selected for anti-schistosomal screening. Summary of calculated physicochemical properties for the 80 compounds: (**a**) hydrogen bond acceptors (HBA); (**b**) hydrogen bond donors (HBD); (**c**) ALogP, and (**d**) polar surface area vs. the molecular weight. Active compounds at 50 µM, but not 10 µM are shown as blue squares. Red squares indicate compounds active at both 10 and 50 µM. Blue circles indicate all inactive compounds.

**Table 1 pharmaceutics-15-01359-t001:** Anti-schistosomal activity summary of the 17 selected compounds.

		EC_50_ (µM) ^1^			
Target Class	Compound Name	Phenotype	95% CI	Motility	95% CI
**HDAC**	**Fimepinostat**	0.447	(0.350 to 0.543)	0.766	(0.256 to 1.275)
	**Trichostatin A**	0.448	(0.177 to 0.719)	0.667	(0.416 to 0.918)
**HSP**	**SNX 2112**	1.799	(1.156 to 2.443)	2.084	(1.732 to 2.436)
	**Radicicol**	2.167	(1.057 to 3.277)	1.489	(0.871 to 2.106)
	**NVP-BEP800**	0.541	(0.154 to 0.929)	0.552	(0.425 to 0.678)
	**Luminespib**	0.197	(0.158 to 0.235)	0.191	(0.152 to 0.230)
**Proteasome**	**Epoxomicin**	0.239	(0.223 to 0.255)	0.212	(0.176 to 0.248)
**Kinase**	**TAE684**	2.886	(2.615 to 3.156)	1.911	(1.801 to 2.011)
	**CGP 60474**	0.301	(0.270 to 0.331)	0.518	(0.389 to 0.648)
	**AZ 191**	4.628	(4.242 to 5.014)	2.177	(1.858 to 2.496)
	**BIO**	2.518	(1.618 to 3.419)	2.163	(1.847 to 2.480)
	**G 5555**	4.433	(3.074 to 5.792)	1.238	(0.406 to 2.071)
	**7070707105**	2.570	(1.598 to 3.542)	2.667	(2.001 to 3.333)
	**MRT67307**	4.736	(3.973 to 5.499)	1.532	(1.224 to 1.840)
	**PHA665752**	5.682	(5.472 to 5.892)	3.345	(2.806 to 3.884)
	**Staurosporine**	0.218	(0.163 to 0.273)	0.187	(0.154 to 0.221)
	**A-674563**	4.124	(3.564 to 4.683)	1.732	(1.127 to 2.337)

^1^ Anti-schistosomula EC_50_ (derived from phenotype and motility metrics) values are calculated based on three dose response titrations (10–0.313 µM). Compounds demonstrating EC_50_ values greater than 4 µM are highlighted in grey, between 1 µM and 4 µM in blue and less than 1 µM in green. 95% CI = 95% confidence interval.

**Table 2 pharmaceutics-15-01359-t002:** ChEMBL-derived cytotoxicity data of the 15 hits across five different cell lines.

	Cell Line (CC_50_, nM)				
**Compound name**	**K56**2	**HCT116**	**A549**	**MCF7**	**Huh-7**
**BIO**	1300	5200	-	-	6200
**A-674563**	-	-	-	-	-
**AZ 191**	-	-	-	-	-
**CGP 60474**	-	-	-	-	-
**Epoxomicin**	-	-	-	-	-
**Fimepinostat**	280 *	5 *	-	41 *	560 *
**G 5555**	-	-	>1000	-	-
**Luminespib**	8.7	121/16	39/60	6.4/3.35	-
**NVP-BEP800**	-	-	-	-	-
**PHA665752**	-	-	-	-	-
**Radicicol**	-	-	100	30/47.7/23/30	-
**SNX 2112**	6	-	-	53	-
**Staurosporine**	40/80/100/76.8	52/37/28/48	7470/9500 (1) 13,220 (2)	8810/520/64	230
**TAE684**	-	-	-	-	-
**Trichostatin A**	410/120/160	800/900	20/160/80/160/50	60/60/100	60

CC_50_ (in nM) indicates the half maximal cytotoxic concentration derived from the literature. When multiple literature data were available, they were all reported. Cell-drug incubation time is 72 h unless otherwise indicated (* = 96 h). “-” when data not available. (1) under normoxic condition; (2) under CoCl_2_-induced hypoxic condition.

**Table 3 pharmaceutics-15-01359-t003:** Cytotoxicity data of three kinase inhibitors on a cancer cell line panel.

	CC_50_ (nM)		
Cell line	CGP 60474	PHA665752	TAE684
COLO-800	940.23	27,169.99	766.95
COLO-684	94.89	-	11,691.29
COLO-824	354.71	-	10,866.86
COLO-829	4176.22	-	28,459.47
COLO-320-HSR	82.86	-	197.76

Data derived from the Genomics of Drug Sensitivity in Cancer project (available from https://www.cancerrxgene.org/). COLO-800 = human melanoma cell line; COLO-684 = Human uterus adenocarcinoma; COLO-824 = human breast squamous cell carcinoma cell line; COLO-829 = Human melanoma cell line; COLO-320-HSR = human colorectal adenocarcinoma.

## Data Availability

These data have been deposited in ChEMBL Neglected Tropical Disease (ChEMBL-NTD) archive. This is available at: https://chembl.gitbook.io/chembl-ntd/downloads/deposited-set-26-cardiff-schistosoma-dataset-3rd-march-2023 (accessed on 3 March 2023).

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
