# Peer review of "Using ChEMBL to Complement Schistosome Drug Discovery"

_pharmaceutics, 2023, doi:10.3390/pharmaceutics15051359_

Round 1

Reviewer 1 Report

General:

The authors present a bioinformatic pipeline aiming to identify potential S. mansoni drug targets within ChEMBL based on homology and available RNAi phenotypes.

The well-written, easy to follow paper is important for the schistosome research community and beyond, as it highlights the need for an integrated science combining the fast and furious advances in genomics/transcriptomics (and proteomics) with the availability of drug repositories and target information in finding new drugs against schistosomes.

Thus, the study shows an interesting way of integrating several schistosome-specific (and non-specific) databases and methodologies to accelerate the identification of new candidate compounds against schistosomiasis.  Although the underlying components are not a breakthrough novelty, the way the pipeline was put together highlights an alternative and bold method for the identification of new drug candidates. 

 Major issues  

  1.  Some references are double-cited throughout the manuscript (e.g. 10 and 11, 75 and 76). Please check carefully
  2. The identified blast targets should be mentioned in a table (supplementary information?).
  3. Adult worm screening was performed with only 1 pair of worms per drug. That is a very low number specially if one is measuring worm pairing stability?
  4. Where is specific information of the targets of the 17 selected compounds (supplementary table?) apart from mentioning in the discussion?
  5. Why not perform ADMET from the start as a one more layer for filtering undesired compounds?
  6. The final 17 hits could have been docked to the schistosoma targets (AlphaFold2 predictions are now widely available on UniProt where no crystal structure exists) to show that a favourable binding pose is achieved. The manuscript would be enriched by showing one or two such poses (optional).

Minor points

Abstract

Line 16: The first sentence in the abstract sounds good but it is  more tunnel of perception than true. There are good and safe drugs against schistosomiasis, vaccinations in advanced clinical trials, and good (albeit not perfect) diagnostics. I am not sure what would make this disease one of ‘humanity’s most recalcitrant NTDs’, compared with e.g. sleeping sickness or leishmaniasis? The authors should consider rephrasing if they agree with my line of thought

Line 26, Sub-molar range. This does not sound very impressive – I think the authors actually mean sub-micromolar?

Introduction

Lines 45-48: “has finally solved both praziquantel and oxamniquine mechanisms of action

“Praziquantel activates a transient receptor potential melastatin ion channel” is  a mechanism of action, but “oxamniquine is bio-transformed into its active form via a sulfotransferase” is not a mechanism of action: it is drug metabolism. Please consider rephrasing.

Line 169: please explain what “HsdOLa:TO - Tuck Ordinary” means and why these outbread mice were chosen, this is not general knowledge

Line 245 (www.microsoft.com) – consider removing seems rather superfluous?

Results

Figure 3 (d) appear to be rather low resolution . Can better images be provided? Also include a scale bar or as a minimum magnification in the Figure Legend

Figure 4 insufficient resolution of chemical structures – please provide higher res images

Lines 329 Grammar: the ex vivo anti-schistosomula potencies of these hits were next explore, potencies is the subject

Figure 5: are these the results of a single experiment, performed once with a single worm pair, or has each experiment been performed at least three times (as lines 211-212 seems to state)? Number of experimental repeats should be stated in the Figure legend

Lines 455-456: “however, we couldn’t identify a single cell line uniformly used for all 15 compounds (Table S4)” This is not a problem, in fact, it would have been a problem to rely on data obtained from a single cell line. Using multiple cell lines actually strengthens the results

Line 543: proteasome, not  proteosome

Line 632: 50 is a bit low, latest count by Roskoski Jr. (PMID: 36403719) is 72.

Reviewer 2 Report

The manusrcipt "Using ChEMBL to complement schistosome drug discovery" by G. Padalino et al. is a very nice paradigm of how a simple bioinformatics pipeline with appropriately selected filtering criteria can be significantly efficient in identifying hit compounds during the early stage of drug development. The authors employed ChEMBL in combination with literature search and selected 80 compounds for screening. Ex vivo testing revealed 50 compounds with activity at 37 μΜ, of which 17 retained their activity at 10 μΜ. This is a remarkably high hit rate.

The experimental work has been carried out appropriately overall, and the analysis and presentation of the results obtained is by far very comprehensive.  The conclusions drawn are supported by the experimental finding, which are nicely presented in the discussion section. As a result, I reckon that this work will be of interest to the readers of Pharmaceutics, and not only those working in the field of anti-schistosomal drug discovery.

Taken together with the high quality of the manuscript, including figures, tables and the full range of the supplementary material provided, I can suggest publication of this manuscript without any hesitation. 

I have only a couple of minor suggestions:

1.  The abstract reads "sub-molar range" in l. 26, but why not "sub-micromolar range" instead?

2. Figure 4 could be more illustrative with larger structures, as in supplementary figure S2.

Reviewer 3 Report

I have no major comments on the manuscript. 

Author Response

Reviewer 3 did not have major comments on the manuscript. 

Reviewer 4 Report

The authors propose the use of  ChEMBL to optimize the schistosome drug discovery pipeline.

The authors proposed using a  ChEMBL-based drug discovery pipeline in the drug discovery process of anti-Schistosome drugs. The manuscript is well-written, and the computational results are experimentally validated. However, there are a few minor concerns:

Abstract: OK

Keywords: please add some more keywords 

A few phrases about molecules used for Schistosoma treatment should be added: their structure, what is the core molecule active in Schistosoma

Materials and methods:

 Why were docking studies not used, at least for the 17 selected compounds, to give further insights about their activity on Schistosoma?

Discussions:

More explicit lines about the structural comparison of the known drugs and the 17 compounds should be added.
